# Prediction of Indoor Air Temperature Using Weather Data and Simple Building Descriptors

**DOI:** 10.3390/ijerph16224349

**Published:** 2019-11-07

**Authors:** José Joaquín Aguilera, Rune Korsholm Andersen, Jørn Toftum

**Affiliations:** International Centre for Indoor Environment and Energy, Technical University of Denmark, 2800 Kongens Lyngby, Denmark; rva@byg.dtu.dk (R.K.A.); jt@byg.dtu.dk (J.T.)

**Keywords:** indoor temperature, machine learning, user feedback, thermal comfort

## Abstract

Non-optimal air temperatures can have serious consequences for human health and productivity. As the climate changes, heatwaves and cold streaks have become more frequent and intense. The ClimApp project aims to develop a smartphone App that provides individualised advice to cope with thermal stress outdoors and indoors. This paper presents a method to predict indoor air temperature to evaluate thermal indoor environments. Two types of input data were used to set up a predictive model: weather data obtained from online weather services and general building attributes to be provided by App users. The method provides discrete predictions of temperature through a decision tree classification algorithm. The data used to train and test the algorithm was obtained from field measurements in seven Danish households and from building simulations considering three different climate regions, ranging from temperate to hot and humid. The results show that the method had an accuracy of 92% (F1-score) when predicting temperatures under previously known conditions (e.g., same household, occupants and climate). However, the performance decreased to 30% under different climate conditions. The approach had the highest performance when predicting the most commonly observed indoor temperatures. The findings suggest that it is possible to develop a straightforward and fairly accurate method for indoor temperature estimation grounded on weather data and simple building attributes.

## 1. Introduction

Exposure to extreme temperature indoors or outdoors may have severe implications for human health. Gasparrini et al. [1] estimated that 7.7% of the total mortality in 384 locations around the globe could be attributed to non-optimum ambient temperatures. In the forthcoming years, climate change will increase the intensity and frequency of extreme weather events, in particular, high temperatures [2]. According to the European Environmental Agency, 80% of the European population is expected to live in urban areas by the year 2020, where health consequences of thermal exposures will be more severe [3,4,5]. However, occupational risks related to thermal exposures of outdoor workers in rural areas may also increase. 

The increasing risk of health impacts from future climate events requires the development of adaptation mechanisms that help people to withstand thermal stressful situations. Heat Shield [6] and ClimApp [7] are ongoing projects that will develop tools for better adaptation to suboptimal thermal environments. ClimApp is a smartphone application that combines climate forecasts, human thermal models, user characteristics and human physiology to improve decision making towards thermal adaptation strategies. Thermal exposure indoors is also considered within the scope of the ClimApp project since it affect the well-being, health and performance of building occupants in particularly fragile populations such as the elderly or young children [8,9]. The estimation of indoor air temperature is essential for the evaluation of thermal comfort and energy consumption in built environments [10,11]. In ClimApp, the estimated temperature will be used as input to the calculation of the Predicted Mean Vote index (PMV). On a seven-point scale ranging from cold to hot, the PMV predicts the average thermal sensation of a group of people exposed to the same thermal conditions [11]. Other input parameters required to calculate PMV are the mean radiant temperature, air humidity, mean air velocity, clothing insulation and metabolic rate. These five parameters will be estimated based on season, geographical location and simple user input. In the future, smartphones with built-in or auxiliary thermal environment sensors may be a useful tool to assess people’s local thermal environment. However, even though most phones are equipped with temperature sensors, they only measure the internal temperature in the phone, which may be heated, e.g., by the phone’s electronics or from lying in a pocket. Technology to measure the thermal environment that occupants are exposed to is not yet fully developed or commonly available [12].

The indoor air temperature depends on multiple factors related with building characteristics, occupants and outdoor climate. Nguyen et al. [13,14] observed that the degree to which indoor and outdoor temperatures are associated depends on the climatic region and that the association is particularly strong during warm seasons. Oreszcyn et al. [15] analysed different factors that influenced indoor temperatures in 1600 low-income English dwellings during the heating season. Their findings indicated that indoor temperatures depended mostly on building characteristics (e.g., construction year and thermal efficiency-related factors, such as insulation level, air tightness of the building envelope) and occupant-related factors (e.g., age and number of occupants). Hamilton et al. [16] found that indoor temperatures in English houses increased with increasing household income and that old houses had significantly lower indoor temperatures than new houses. French et al. [17] analysed temperature data from 400 houses in New Zealand. Their results showed that heating type, climate and house age had the largest influence on winter indoor temperatures, whereas the availability of air conditioning, house age and outdoor climate influenced temperature levels more during summer. Magalhães et al. [18] also analysed the factors that influence indoor temperatures using enhanced linear regression models with measurements from field studies in 141 dwellings. Their findings showed that the variability of daily mean indoor air temperatures was influenced by building characteristics (73% to 85% of the surveyed households), socio-economic factors (4% to 14% of the surveyed households) and outdoor air temperatures (1% to 3% of the surveyed households). Building characteristics included parameters such as age of construction, wall insulation, window characteristics and type of space heating equipment. Moreover, their method to estimate indoor temperature showed a high predictive performance (R^2^ from 0.89 to 0.91) when comparing predicted and measured temperature values. White-Newsome et al. [19] applied mixed linear regression models to predict hourly indoor temperature measurements during summer in 30 residences. The combination of outdoor temperature, solar radiation and dew-point temperature explained 38% of the variability in the indoor temperature values in their study. Several studies developed more advanced techniques for indoor temperature prediction than merely linear regression models. Some of them used Time-Series analysis [20] and others used a combination between Time-Series and Artificial Neural Networks (ANN) [21,22,23,24]. Time-Series is an approach used for data forecasting based on statistical analysis of measured values over a defined period. Such a method is normally used for Model Predictive Control (MPC), applied in modern Heating, Ventilation and Air-Conditioning (HVAC) operation strategies. Mateo et al. [25] applied different machine learning techniques to forecast indoor air temperatures based on outdoor climate parameters (air temperature and relative humidity) and variables related to space heating use (temperature set point and heating power). Their approach focused on short-term temperature forecasting (24 h), reaching an average error of approximately 0.1 °C. The framework proposed by Kelly et al. [26] was able to predict indoor air temperatures with an error of 0.71 °C at 95% confidence. Their method used behavioural, environmental and building efficiency variables as inputs, which were processed through panel Time-Series methods. In general terms, the outcome of Time-Series forecasting is restricted to the specific design and operation conditions of a building. Its complexity prevents its implementation in smartphone applications. 

The framework suggested in this study uses occupants’ observations and weather data as input to estimate indoor air temperatures. It is expected that the indoor module of the App will be particularly useful in environments hosting fragile individuals, such as young children and the elderly, and where heat or cold spells may result in unusual thermal exposures. Under these conditions, the App’s output may assist in an evaluation of coping actions. All input parameters were combined through a decision tree algorithm which was constructed based on measured data and building characteristics obtained from field studies. The framework was developed to be integrated into ClimApp to enable the assessment of thermal exposures indoors by calculating the Predicted Mean Vote based on weather data and simple building descriptors. The scope of this paper is limited to the development of a prediction framework for indoor air temperature.

## 2. Materials and Methods 

Weather data and building-related parameters were obtained from field studies to determine to which degree they are related to measured air temperatures (TA). As presented in Figure 1, the first step was to gather data from field studies. Local weather data was obtained from weather forecast agencies available online, whereas building-related parameters were collected through questionnaires given to the occupants of each building. This information will normally be provided through the ClimApp interface based on what occupants know about the building and what they can observe from their surroundings. Then, the input data was processed and used to train a decision tree model. The performance of the model to predict TA values correctly was tested based on the data collected from field studies. In addition, a building simulation model was developed, which was used to evaluate how generalizable the framework was in predicting TA in different climate conditions. A parametric analysis was then carried out to evaluate which variables contributed to the predictive power of the algorithm. In practical use after implementation in the app, correct estimation of TA depends to a high degree on the quality of the feedback provided by the users of the app.

### 2.1. Data Collection

The input data used to construct and test the framework presented in this paper was obtained from measurements performed in seven dwellings located in Copenhagen, Denmark. This database was originally obtained through field studies by Andersen et al. [27] and Fabi et al. [28]. Indoor environment parameters (temperature, relative humidity and CO_2_ concentration), outdoor climate (air temperature (TAO), relative humidity (RHO) and solar radiation (SR)) and parameters related to occupant behaviour were measured and monitored across different seasons during the period from March to August 2008. As presented in Table 1, the dwellings analysed in the field study had either natural or mechanical ventilation. The heating system of all the residencies was based on water radiators connected to a district heating network and none of them had mechanical cooling systems. In each dwelling, the measurements were taken either from the master bedroom (BR), from the living room (LR) or from both rooms.

All the variables presented in Table 2 were considered as input parameters to construct the algorithm, except the indoor CO_2_ concentration, which was used to estimate whether the room was occupied or not. The hourly running mean outdoor temperature (TRM) was calculated as the exponentially weighted moving average, considering the previous values of mean hourly outdoor air temperatures (Equation (1)).
(1)$$TRM_{t}=\left(1-\alpha \right)\left[{\bar{TAO}}_{t-1}+\alpha {\bar{TAO}}_{t-2}+\alpha^{2}{\bar{TAO}}_{t-3}+\ldots \right]$$
where $TRM_{t}$ is the running mean outdoor temperature for an instant *t*; ${\bar{TAO}}_{t-1}$, ${\bar{TAO}}_{t-2}$ and ${\bar{TAO}}_{t-3}$ are the mean hourly outdoor temperatures for the previous hours; and *α* is a constant between 0 and 1 that defines the speed at which the running mean reacts to variations of the outdoor temperatures. The TRM was finally calculated using the three previous values of hourly outdoor temperature with a coefficient *α* equal to 0.9, since the correlation between TA and TRM reached its highest value (Pearson coefficient = 0.14) under such conditions. Outdoor environment parameters (TAO, RHO and SR) were obtained from the meteorological station closest to the dwellings [29]. Indoor climate parameters (TA, CO_2_) were measured using Hobo U12-012 data loggers (Onset Computer Corporation, Bourne, USA) connected to Vaisala GMW22 CO_2_ transmitters (Vaisala Corporation, Helsinki, Finland). Window opening (WO) was measured using HOBO UA-004-64 Pendant G accelerometers (Onset Computer Corporation, Bourne, USA). The heating set point was measured on the thermostatic radiator valves (TRV) connected to the Hobo U12-012 data loggers [30]. The value of the TRV could be adjusted between −1 and 6, depending on the desired set point (from 7 to 30 °C).

### 2.2. Data Processing

Indoor climate sensors were placed on internal walls with a minimum distance to the closest radiator of one meter and at a height of approximately 1.8 m above the floor. The temperature sensor was placed inside the casing of the Hobo U12-012 data logger which could have been heated by direct solar radiation. Placing the temperature sensors in shaded areas was intended so that they were not affected by direct sunlight. However, this was not possible in all cases, due to practicalities and the acceptance of the residents. Consequently, the indoor temperature measurements were corrected when direct sunlight hit the sensors by applying a linear interpolation of the temperature measurements between 30 min and one hour after the illuminance level measured on the sensors was larger than 1000 lux. 

Only measurements during occupied hours were considered in the framework. The CO_2_ concentration level was used as an indicator of occupancy in each room, considering that occupants were the main source of CO_2_ indoors [31]. It was assumed that a room was unoccupied when the CO_2_ level was equal to or below 430 ppm. This value was observed to correspond to the average outdoor CO_2_ concentration plus the sensors’ uncertainty in the CO_2_ measurement. According to Andersen et al. [27], one of the most common reasons for closing windows is people leaving their dwelling. Therefore, it was considered that a room was also unoccupied when the windows were closed and the CO_2_ concentration decreased continuously until a level of 430 ppm or below. The occupancy level was determined as a binary variable (occupied or not occupied) as it was not possible to verify the exact number of occupants present in a room within a defined period.

### 2.3. Model Construction

To estimate TA, it was necessary to find its relationship with discrete (WO, NO and CY) and continuous (TRM, RHO, TRV, FA and SR) variables that could influence its value. The problem was solved with a classification model which could handle both discrete and continuous variables. TA values were converted into a pre-defined number of categories, depending on the desired precision of the prediction. In this study, the measured TA values were between 16 °C and 30 °C. As presented in Table 3, temperatures were divided into seven categories that each account for an interval of ±1 °C. In other words, the outcome of this method was the prediction of a temperature interval where the actual TA value is included. According to Liu et al. [32], the discretization of a continuous variable simplifies the construction of a rule-based algorithm, making the predictions more understandable and often leading to a higher predictive accuracy. The method developed in this study was thought to be implemented in a smartphone App. Thus, it was essential to as simple and reliable as possible. 

A C4.5 decision tree algorithm was the tool used to predict TA [33]. A decision tree is an algorithm that makes predictions by calculating the probability of an outcome to occur, based on attributes that influence it. It is a tree-shaped algorithm, where each node represents an attribute, a branch corresponds to a decision rule and the leafs are the possible outcomes [34]. The C4.5 decision tree was implemented in Java using WEKA (version 3.9, University of Waikato, Hamilton, New Zealand) as a machine learning workbench [35]. 

The learning method used by the C4.5 algorithm is based on the concept of divide-and-conquer, whereby a certain attribute *X* divides the training data set of *T* into *n* subsets $T_{1},\;T_{2},\;\ldots ,T_{n}\;$, where each $T_{i}$ is a subset of instances. In this study, an instance or data point corresponded to a discrete group of attributes that was aligned with a matching TA measurement (e.g., a data point composed by the following attributes: WO = 1, RHO = 50, SR = 0, TRV = 1, TRM = 16, CY = 1954, FA = 24 and NO = 2, corresponded to a TA inside class *c*). The algorithm then used the concepts of Information Entropy H and Information Gain to define the relevance of each attribute to estimate each class *c*. Information Entropy is used to measure the homogeneity of a sample distribution, which corresponds to the amount of information needed to identify a class. For this, it is needed to determine $p_{T_{i},c}$ that is the proportion of instances of $T_{i}$ that belong to a class *c* (Equation (2)). Then, the Information Gain is applied to calculate how much an attribute contributes to estimate *c*, which is defined as the change in entropy (Equation (3)) [34].
(2)$$H\left(T_{i}\right)=-\overset{C_{T}}{\underset{c=1}{\sum }}p_{T_{i},c}\cdot log_{2}\left(p_{T_{i},c}\right)$$
(3)$$Information\;Gain\left(X,T\right)=H\left(T\right)-\overset{n}{\underset{i=1}{\sum }}\frac{\left|T_{i}\right|}{\left|T\right|}\cdot H\left(T_{i}\right)$$
where $p_{T_{i},c}$ is the proportion of data points or instances belonging to a class *c*, *C_T_* is the total number of classes and $T_{i}$ is one sample among all the *n* subsets in which the total amount of training data *T* was divided due to an attribute *X*. Considering a group of data points or instances, the algorithm grows an initial tree evaluating which is the attribute that diminishes more the entropy of the partition. Each of the partitions is treated again as a new tree, repeating the process until there are no misclassifications. 

### 2.4. Building Simulation Model

It is challenging to obtain datasets across a wide variety of climates that contain indoor and outdoor thermal environmental measurements as well as HVAC operation parameters in real applications. To overcome this constrain, a comprehensive dataset was generated through a dynamic building simulation model implemented in IDA-ICE (version 4.8, EQUA Simulation AB, Stockholm, Sweden), which is a simulation tool that models a building, its systems and their control [36]. This allowed the evaluation of the potential generalization of the framework presented in Section 2.3. Three different locations were considered for the simulations: Copenhagen in Denmark (55.62 °N, 12.65 °E), Athens in Greece (37.9 °N, 23.73 °E) and Abu Dhabi in the United Arab Emirates (24.4 °N, 54.5 °E). Those locations were chosen since they account for climates with different air temperature, relative humidity and solar radiation levels, as shown in Table 4. The meteorological data used in the simulations were obtained from ASHRAE’s IWEC (International Weather for Energy Calculations) [37]. 

The model corresponded to a single-family house with a master bedroom (BR) and a kitchen-living room (LR). The main characteristics of the model are presented in Appendix A. Considering the age of the houses analysed in the field study (see Table 1), the envelope characteristics of the model (see Table A1 and Table A2 in Appendix A) accounted for a typical single-family Danish house constructed in the period 1951–1960 [38]. The models were mechanically heated with ideal heaters (heating components without thermal inertia) and cooled through window opening. The dwelling in Abu Dhabi also had mechanical cooling to avoid overheating, implemented as a mixed-mode cooling system [39]. The control approach for window opening and the heating/cooling system was based on the Adaptive Thermal Comfort model, which determines the indoor comfort temperature based on outdoor climate [40]. The set point to turn on/off water radiators corresponded to the lower limit of the Adaptive regression model presented in EN15251 [41] for an indoor climate Category II. Windows were opened based on the upper limit of the same model minus 0.5 °C, whereas the set point for mechanical cooling used the same upper limit plus 0.5 °C. All control signals were determined by proportional controllers with a proportional band of 1 °C. As a result, the operation of the control system for window opening was able to open windows when the indoor temperature was lower than the upper limit of the adaptive model. When the indoor temperature was over that limit, the windows were closed and mechanical cooling was applied. This approach aimed to simulate occupants opening windows to maintain the indoor temperature within their thermal comfort limits. Mechanical cooling was then used when cooling by the opening of windows was not enough to maintain occupants’ comfort levels.

The data obtained from the simulations were implemented to provide input data for a model that determines TA based on the prevalent outdoor climate, building-related parameters and occupancy level.

### 2.5. Performance Evaluation

The performance of the method was evaluated based on the F1-score of a multi-class classifier [42]. This performance indicator was calculated based on the precision and recall indicators. Precision represents the proportion of instances correctly classified as positives (Equation (4)), whereas recall corresponds to the effectiveness of an algorithm to identify positives (Equation (5)). In this study, a positive refers to a particular TA class that is being predicted and a negative corresponds to the rest of the classes. Considering the task of predicting a TA value in class A, a high precision means that the method is able to correctly predict that such TA is in class A out of all the predictions of A, while a high recall reveals that is also able to correctly predict that a TA is in A out of all the correct predictions. According to Kautz et al. [43], the outcome of those indicators can be misleading with imbalanced data, i.e., categories with a different number of data records. The number of TA measurements in each of the classes in Table 3 was expected to differ, leading to an uneven distribution of data. The F1-score is a measure of a test’s accuracy and it was used to account for the imbalance. However, unlike a simple accuracy calculation (number of correct predictions over the total number of predictions), the F1-score requires a similar number of correct predictions across all the classes to reach a value close to 100%. It is calculated as the harmonic mean of precision and recall (Equation (6)).
(4)$$Precision=\frac{\sum_{i=1}^{l}tp_{i}}{\sum_{i=1}^{l}(tp_{i}+fp_{i})}$$
(5)$$Recall=\frac{\sum_{i=1}^{l}tp_{i}}{\sum_{i=1}^{l}(tp_{i}+fn_{i})}$$
(6)$$F1\;score=2\cdot \frac{precision\cdot recall}{precision+recall}$$
*l* corresponds to the total number of classes; $tp_{i}$ is the number of true positives per class *i*; $tn_{i}$ is the number of true negatives per class *i*; $fp_{i}$ is the number of false positives per class *i*; $fn_{i}$ is the number of false negatives per class *i*.

Two approaches were applied to evaluate the performance of the algorithm. The first approach, named Validation, randomly mixed the data obtained from all dwellings and then divided it for training and testing. In this method, a 10-fold cross-validation was used, which comprised the division of the data set into 10 smaller groups or folds with an equal number of data points. Nine folds were used to train the algorithm and one was used to test it. This process was repeated 10 times, averaging the F1-score of each step. The second method, called Application, used the data obtained from six dwellings to train the algorithm and testing it by using the TA measurements from the seventh dwelling, repeating this process seven times (each one considering a different dwelling to test the algorithm). This concept was used to evaluate the performance of the framework using an unbiased data set by testing it with data from a completely different building. 

## 3. Results 

This section presents the main results of the analysis. Table 5 shows all the measured variables in each of the bedrooms and living rooms. All the TA measurements were between 17 °C and 30 °C, with the median between 19 °C and 24 °C depending on the dwelling. The TRV values showed a low variability in some dwellings (D3-LR, D4-BR, D5-BR and D11-LR). However, this did not necessarily cause a lower variability in the TA measurements in those dwellings. 

Figure 2 shows that the measured TA values were distributed differently among the temperature categories in Table 3. Categories with the lowest and highest TA values (e.g., the intervals 16 ≤ TA < 18 and 28 ≤ TA ≤ 30), had a lower number of data points compared to the other categories. Moreover, the distribution of TA values shows that not all the classes were considered in the data sets from all dwellings.

### 3.1. Attribute Selection

The C4.5 algorithm applied in this study uses the concept of Information Gain to select the most important attributes affecting TA. Hence, Information Gain (Equation (3)) was calculated for each input parameter to assess its importance for the construction of the set of rules of the decision tree. Moreover, the correlation between the input parameters and their corresponding TA values was evaluated through Pearson’s correlation coefficient. The results presented in Table 6 show that TRV was the input parameter that added most to the prediction, as it had the highest Information Gain. Attributes with higher correlation coefficients were not necessarily those that provided more information to develop the algorithm.

### 3.2. Performance Evaluation

The results from the Validation approach showed that the performance of the algorithm had its highest increment when the number of data points used for training was below 10,000 (Figure 3). The precision and the recall of the method had similar values regardless of the amount of training data. The similarity between both indicators and the high F1-score achieved (92%) suggests that the method was able to classify the TA values with a similar performance for all the TA classes. This was a result of the equivalent distribution of the training and testing data, as both were extracted from the same data set. Figure 3 shows that the maximum performance was reached with 95,000 training data points. Considering that each data point corresponded to different parameters measured every 10 minutes, 1000 data points was equivalent to 6.9 days of continuous measurements. Hence, the method required 3.3 days of continuous measurements to have a predicting performance of at least 50%.

The Application method was applied with the same amount of training data and the same amount of testing data in each evaluation. The size of the data set used for testing (7131 data points) corresponded to the minimum number of measurements in a single dwelling among all seven dwellings. The minimum number of measurements in the remaining six dwellings corresponded to the size of the training data set (74,582 data points). The results show that the Recall outweighs the Precision for all the tested cases (Figure 4). In other words, the algorithm is better at predicting certain TA values correctly, but not all of them. This is due to an uneven distribution among the different classes presented in Table 3. The lack of correct predictions for categories that contained fewer data points affected the overall Precision of the method and therefore, the F1-score.

The results from Figure 5 show that the method was not able to estimate the TA included in categories A and G correctly, corresponding to the temperatures inside the intervals 16 ≤ TA < 18 and 28 ≤ TA ≤ 30, respectively. When comparing these results with those from Figure 1, it can be seen that the method had a higher performance when predicting the most prevalent TA values.

### 3.3. Building Simulation Results

The data from the field experiment were applied to train the TA-prediction method, whereas the data obtained from the building simulation model were used as testing data. Four additional TA classes were added to the method to account for the higher TA values obtained from the simulations from Athens and Abu Dhabi. Such classes considered the following TA intervals: 30 ≤ TA < 32, 32 ≤ TA < 34, 34 ≤ TA < 36 and 36 ≤ TA ≤ 38, which only applied to the data obtained from the two locations previously mentioned. The results presented in Table 7 show that the F1-score differed depending on the location and the input parameters taken into account. The performance of the method using real data corresponded to a parametric evaluation using the average results from the Application analysis presented in Section 3.2. Overall, the method showed a lower performance when predicting the simulation data set than for the real data. Within the simulation results, the F1-score was the highest for Athens, compared to the results using the data from Copenhagen and Abu Dhabi. The performance decreased when some of the attributes were not considered as input parameters (highlighted values in Table 7), which depended on the climate considered. However, the absence of SR, TRM and NO decreased the performance of the algorithm when using the testing data from Athens and Copenhagen. However, only the omission of RHO and SR had a negative impact on the performance when the data from Abu Dhabi were used for testing. The presence of mechanical cooling in the model used for that location probably influenced the prediction of the TA values, which was also reflected in the lower performance achieved in that scenario (13.3% for the optimal case). When no mechanical cooling is taken into account, SR, TRM, and NO can be considered as the necessary inputs that the TA-method requires in order to reach the highest overall performance. Other additional attributes such as the cooling set point could be included as input when a building is mechanically cooled. 

Unlike the results obtained from the Information Gain analysis, the TRV and FA did not contribute to the accuracy of the method tested on the simulation model data. The reason was that the simulation model included data for a full year rather than only for six months as the field study data. This had an effect on the influence on TA of the TRV, which expectedly was higher during the heating season. Moreover, the data from the field experiment were acquired from buildings with different dimensions and the data from the simulation model were obtained from only one building type, decreasing the importance of the floor area (FA) as the input parameter in that case.

Figure 6 shows that the prediction reached a maximum performance of 68% with the most frequent TA values from Athens, whereas it had a more modest performance of 50% when predicting the data from Copenhagen and Abu Dhabi. As mentioned in Section 2.5, the F1-score takes into account the number of incorrectly predicted TA classes as well as the correctly predicted ones. Thus, the results show that the method is able to correctly predict the most frequent TA temperatures with a probability of 68% and an uncertainty of ±1 °C. The distribution of the TA values in the testing data was within a broader temperatures range for Athens (from 16 °C to 38 °C) and a higher temperature level for Abu Dhabi (from 24 °C to 38 °C) than for Copenhagen (16 °C to 30 °C). The results from the building simulation evaluation show that the highest performance was achieved when predicting the most populated TA classes from the training data, which corresponded to temperatures between 16 °C and 30 °C (see Figure 2). Hence, the method requires that the TA values in the training data are distributed as evenly as possible across a broad range of temperatures. This maximizes the performance of the method under diverse climate conditions.

### 3.4. Estimation of the Predicted Mean Vote

This study focused on the development of a TA-prediction approach to assess thermal indoor environments based on the Predicted Mean Vote (PMV). In practice, the App users will be presented with a visual indicator based on the PMV rather than predicted TA values. Table 8 shows that the prediction of TA in intervals of ±1 °C resulted in estimation of the PMV with an overall uncertainty of ±0.2 and ±0.3 for clothing levels of 1 clo and 0.5 clo, respectively. The uncertainty level was calculated as the mean distance between the maximum and minimum PMV within the estimated TA interval. The estimation of the PMV was based on a relative humidity of 50%, an air velocity of 0.1 m/s, a metabolic rate of 1.2 met and a mean radiant temperature equal to the room air temperature. The results from Table 8 show that the accuracy (F1-score) of estimating the PMV depended on how many TA values from the testing data set were within an interval. This value was different for the data from Copenhagen, Athens and Abu Dhabi and yielded maximum accuracies of 50%, 68% and 49% at these locations.

## 4. Discussion

As expected, the measured TA values were not evenly distributed among the different temperature classes. In particular, none of the dwellings except D3 provided TA values inside all classes. As described by Liu et al. [32], when a binning discretization method is used (division of a continuous attribute into a specified number of bins), there is a trade-off between creating classes with equal frequency and creating classes with equal width. In this study, discretizing the TA values comprised the creation of temperature ranges with equal width to define beforehand an estimation of the uncertainty of every TA value predicted, corresponding to ±1 °C. This definition caused an uneven distribution of the data records used to develop the classification algorithm, but it may have allowed predicting the PMV with reasonable accuracy.

The outcome of the Validation evaluation shows that the algorithm had a performance of 92%, estimated based on its precision, recall and F1-score. However, the training data and validation data were separate data sets extracted from the same group of measured values. Therefore, the distribution of the training data and the data used for validation was similar, which explains the similarity between recall and precision. The Validation evaluation showed that the algorithm’s performance is probably high when the testing conditions are taken into consideration during the training phase. This means that the magnitude of variables, such as type of building, outdoor climate, heating system and occupant behaviour were similar during the training and testing phases. However, such type of evaluation did not estimate how generalizable the TA-prediction method may be. The Application evaluation showed that the performance of the algorithm decreased significantly when tested in a completely different dwelling (Figure 4). The prediction performance was affected by an uneven distribution of TA values across the predicted classes, producing differences between the obtained recall and precision. The results show that the method presented in this study had a higher probability of predicting correctly the most prevalent temperatures in the different dwellings. The discretization of the TA measurements prevented the method from correct estimations in temperature categories lacking data points. This class imbalance is unavoidable, given that in reality, it is not possible to have equal number of measured TA values during equal periods. Predicting incorrectly the most extreme temperature values implies that the method will not be able to assess the most critical situations when occupants are under unacceptable thermal environments (e.g., inactive heating system during winter, windows opened during cold outdoor temperatures). However, one evident solution to increase the prediction performance of the excessively high or low indoor temperatures is to use training data from a longer period. As suggested by Japkowicz and Stephen [44], increasing the size of the training set decreases the effect of the class imbalance, improving the performance of the method. For that, the training set should account for variations of TA values during different seasons, where the heating and cooling needs change. 

The initial criteria to select the input parameters was to include variables that can be obtained from weather services online and from relatively simple user-provided feedback. The literature review presented in the introduction showed that building-related parameters, outdoor climate and occupant behaviour had an effect on indoor temperature values in several studies [13,14,15,16,17,18,19,20,21,22,23,24,25,26]. A comprehensive database extracted from field studies [27,28] was used to train and test the method. This database contained building descriptors (floor area and construction year) and occupant behaviour (heating set point, window opening and nominal occupancy level). However, since the database was obtained only from one climatic zone (Copenhagen, Denmark), the method was tested under different climate conditions using dynamic simulations. Even though eight parameters were used as inputs, the results from Table 7 show that not all of them had the same impact on the performance of the method. Depending on the parameters considered, using more descriptors has the potential to improve the prediction performance. Nonetheless, this will probably make the task of getting input from occupants more difficult and may induce over-fitting (a predictive model that is highly specific to only one set of data [45]).

Based on the building simulation analysis, it was found that the outdoor climate affected the performance of the TA-prediction method, which also had an impact on building-related attributes used as input parameters. The method had its highest performance (68%) when predicting the most frequent TA values. Hence, the training data should account for a wide range of temperatures evenly distributed among the TA prediction classes. When the method was applied to the data from Copenhagen and Athens, three attributes were relevant as input to predict TA (solar radiation, hourly running mean temperature, and number of occupants), whereas only two variables (relative humidity outdoors and solar radiation) were important when using the data from Abu Dhabi. The simulations considered the same building type for Copenhagen, Abu Dhabi and Athens using Danish construction regulations. According to the studies from Böhnke [46] and Giusti and Almoosawi [47], a representative dwelling for Athens and for Abu Dhabi is less tight with significantly higher heat losses than a Danish household. Nevertheless, the building type was not changed in the simulations performed in this study to better analyse the influence of climate-related parameters on the performance of the method. 

The hourly running mean outdoor temperature was calculated based on past hourly values of outdoor temperature. Due to the thermal mass of a building, the outdoor air temperature does not have an instant effect on the air temperature indoors, which explains the importance of TRM as an input parameter to estimate TA. Vant-Hull et al. [48] observed that hourly averages of indoor and outdoor temperatures were correlated with a time lag of 2 h, based on a study performed in 30 residences located in New York City, USA. Their results differed with the period of 3 h considered to calculate the TRM in this study, probably due to discrepancies between the thermal mass of the buildings analysed in both studies. The results were in agreement with the findings from Nguyen et al. [14] and French et al. [17], since the TA predictions depended on outdoor climate variables (TRM and RHO) during the warm season. Building-related parameters, such as CY and FA, only had a modest influence on the prediction performance, regardless of how well they correlated with the TA values and their Information Gain measure. The results were not entirely in accordance with the studies by Oreszyn et al. [15], which observed that TA values in households were influenced by building age. Their study was carried out during wintertime, when the building envelope characteristics have a more noticeable effect on the temperature indoors. The field study presented in this paper was not restricted to a single season as it was performed during winter, spring and summer. Therefore, the TA values were probably less dependent on the envelope of the buildings analysed. Moreover, the study by Kragh and Wittchen [38] showed that the overall heat transfer coefficient of Danish dwellings did not change significantly during the years that the households analysed in this study were originally constructed (between 1928 and 1981). However, Oreszyn et al. [15] found that the number of occupants was a factor that significantly affected the TA values, which is in accordance with the results from this study. The presence of humans in indoor environments increases the heat load and therefore, TA, which varies depending on the number of occupants and most likely, their metabolic activity rate. The parametric analysis in the current study only analysed the absence of one attribute at a time, evaluating its influence on the overall performance of the method. In practice, more than one parameter can be missing or some of them can be incorrectly provided by the App user. This type of issue will probably have a greater effect when predicting TA values, which were not possible to estimate in this study.

The results suggest that the relationships between the different parameters used in the model and TA are context-dependent. Diverse heating/cooling systems may have a larger or smaller influence on TA (i.e., higher or lower influence of TRV), which could be caused by technologies using different heat transfer principles or because of different climates. Furthermore, occupants’ control possibilities over their thermal environment could be reduced or even non-existent in some cases, whereas in other buildings with personal control systems, individuals are able to control their immediate surroundings. Even though the value of TA relies on the context, it is evident that the outdoor climate parameters have multiple direct or indirect possibilities to influence the TA values. Namely, they have an effect over human behaviour, over the performance of heating/cooling systems and a direct influence over TA. A TA-prediction model is not capable of generalizing such relationships for all different circumstances (e.g., different building types, climates, human behaviour). Nevertheless, the framework presented in this study is able to learn from those circumstances, making meaningful predictions of TA.

The main outcome of this study was a method to predict indoor air temperature to estimate PMV and assess indoor thermal comfort with incomplete knowledge of the real thermal conditions. When the method has been implemented in the app, the user may enter specific information on the building and behaviour related parameters, such as the state of thermostats or window opening, which will increase the accuracy of the predicted exposure. The user may also be unaware of these building descriptors, in which case default values will be used as input to the PMV prediction. Thus, in addition to the uncertainty of the predicted indoor air temperature, several other factors will affect the accuracy of the estimated PMV. Therefore, a conservative approach should be used when users apply the app to assess thermal strain indoors. This can be done by introduction of a safety margin, e.g., corresponding to the uncertainty shown in Table 8. Under non-comfort conditions, the predicted air temperature may also be used to calculate other indices, typically with more extreme thermal exposures that are more common outdoors, such as the wet-bulb globe temperature (WBGT), predicted heat strain (PHS), or required clothing insulation (IREQ) [49,50,51].These other indices are already used in the app to assess outdoor thermal exposure. After implementation of the suggested method in the app, its functionality will be tested in a range of field studies as part of the ClimApp project.

The framework presented in this study neglects the time-dependent nature of the parameters used to develop the model using a classification approach rather than a time-series regression model to predict indoor air temperatures. Moreover, it applies univariate discretization only based on temperatures, which inevitably also divides the input parameters into categories, affecting its prediction performance. However, the application of a more complex discretization approach or a regression model that accounts for time-dependent variations will increase the need for more computing power. The simplicity of the method proposed in this paper is grounded in the possibility of its application into mobile devices, giving a meaningful thermal evaluation of users’ indoor environments. Additionally, this method could be used in future research studies to develop techniques to monitor indoor environments with fewer or no sensors. Tronchin et al. [52] highlighted the importance of limiting the number of sensors in buildings to increase the possibility of monitoring at multiple scales.

The generalization of the TA-prediction model depends on its ability to produce correct predictions under different conditions (e.g., building type, climate, occupants). The measured data considered in this study accounted for only one climate and one building code. The simulated data had a wider geographical scope to include different climates and improve the generalizability of the prediction. However, more research efforts should be made to develop a comprehensive database that allows the analysis of building performance, indoor environment and occupant behavior under an even wider range of conditions and improves the prediction power of data-driven models.

## 5. Conclusions

This paper proposed a method to predict indoor air temperatures based on weather data and simple building descriptors, which are obtained from users of the method. Eight input parameters were tested: three outdoor climate parameters (outdoor running mean temperature, outdoor relative humidity and solar irradiation), three building-related parameters (floor area, number of occupants and construction year) and two parameters related with occupant behaviour (thermostat setting and window opening). The method was analysed based on data from seven Danish dwellings obtained during a measuring period of six months. Building simulations were used to test the method under different climate regions due to the lack of comprehensive datasets from other climates. The method had an accuracy of 92% (F1-score) to predict indoor air temperatures with an error of ±1 °C when tested under previously known conditions (e.g., same building type and occupant behaviour). The accuracy for the same prediction was only 30% when evaluated under completely new conditions under the same climate and it decreased when tested in new climates. The performance of the method was affected by considering a discretization of indoor air temperatures and by only applying measured data from a single climate for its construction. However, the method was able to correctly predict approximately 68% of the most frequent temperature levels. Solar irradiation, outdoor running mean temperature and number of occupants were the parameters that were most important and increased the accuracy of the predictions of indoor air temperature, whereas building related parameters (construction year and floor area) only had a minor influence. Finally, the outcome of this study shows that it is possible to develop a simple method that predicts indoor air temperature with reasonable accuracy based only on weather data and occupant-provided feedback.

## Figures and Tables

**Figure 1 ijerph-16-04349-f001:**
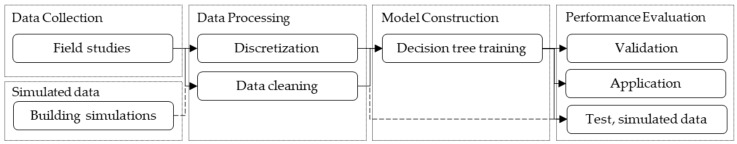
Process map of the methodology applied in this study. The data collected from field studies is represented with a solid line, whereas the simulated data is illustrated with a dashed line.

**Figure 2 ijerph-16-04349-f002:**
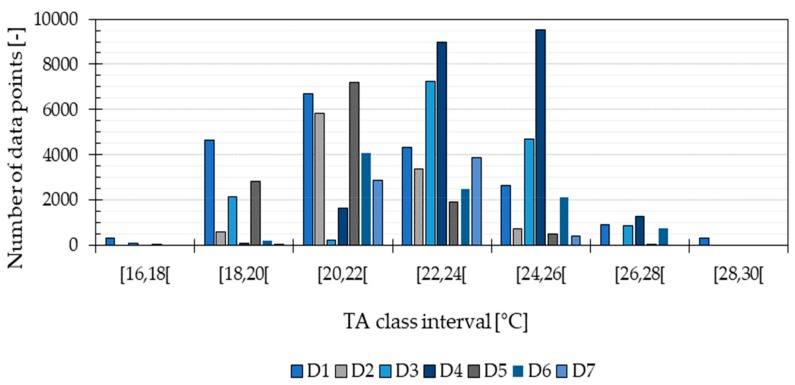
Number of data points in each TA class obtained from all the dwellings (D1, D2, D3, D4, D5, D6 and D7), considering BR and LR together.

**Figure 3 ijerph-16-04349-f003:**
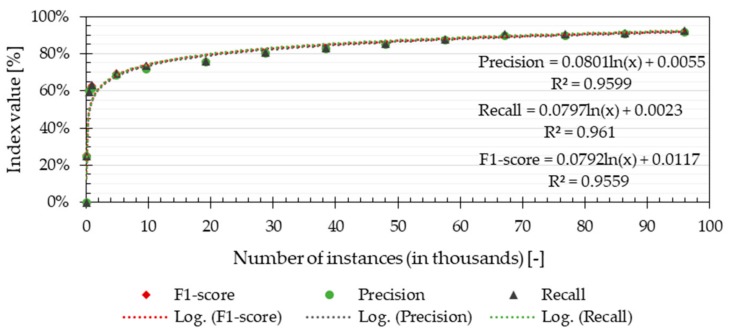
Overall performance of the prediction algorithm using the Validation approach as a function of the number of data points used as training set.

**Figure 4 ijerph-16-04349-f004:**
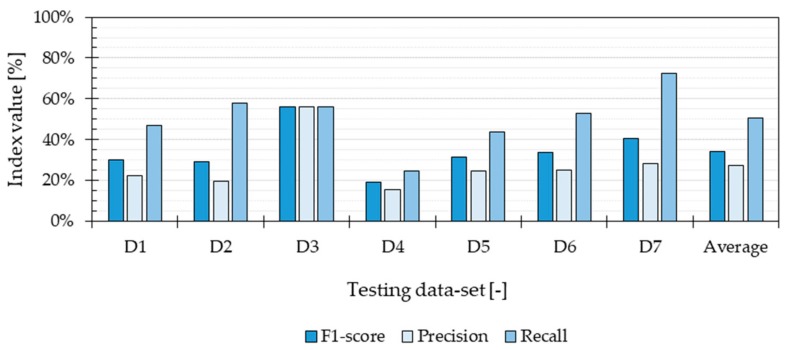
Overall performance of the method when the Application approach was used, which accounts for the data from the seven dwellings (D1, D2, D3, D4, D5, D6 and D7) evaluated in the field study.

**Figure 5 ijerph-16-04349-f005:**
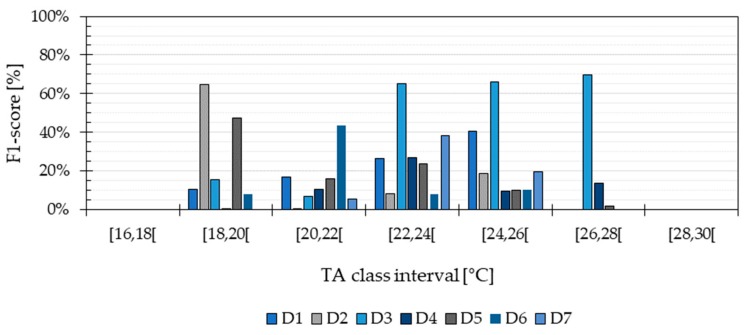
F1-score for all seven TA classes when the Application approach was used, which accounts for the data from the seven dwellings (D1, D2, D3, D4, D5, D6 and D7) evaluated in the field study.

**Figure 6 ijerph-16-04349-f006:**
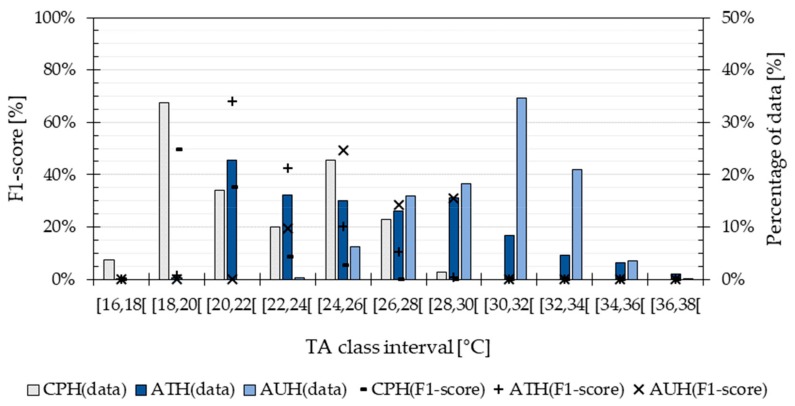
Distribution of the TA values and performance of the TA-prediction method when forecasting data from the building simulation model located in Athens (ATH), Copenhagen (CPH) and Abu Dhabi (AUH).

**Table 1 ijerph-16-04349-t001:** Characteristics of the dwellings investigated related. LR: living room, BR: bed room.

Dwelling Index	Number of Residents	Floor Area (m^2^)	Ventilation Type	Construction Year (Renovation)	Closest Meteorological Station (km)	Room Measured
1	2	145	Natural	1928	13	LR,BR
2	2	130	Natural	1956 (1976)	4	BR
3	2	83	Mechanical	1981 (2001)	11	LR,BR
4	2	86	Natural	1945	5	LR,BR
5	1	83	Mechanical	1981 (2001)	11	LR
6	3	87	Natural	1945	5	LR,BR
7	2	77	Natural	1945	5	LR

**Table 2 ijerph-16-04349-t002:** List of attributes used to train air temperature (TA)-prediction method.

Variable	Type	Unit
Indoor air temperature, TA	Continuous measurement	°C
Running mean outdoor temperature, TRM	Continuous parameter	°C
Outdoor relative humidity, RHO	Continuous measurement	%
Global solar radiation, SR	Continuous measurement	W/m^2^
Position of the heating set point, TRV	Continuous measurement from −1 to 6	-
Window opening, WO	Discrete value: open = 1, closed = 0	-
Indoor CO_2_ concentration, CO_2_	Continuous measurement	ppm
Floor area of the room, FA	Continuous parameter	m^2^
Construction year, CY	Continuous parameter	years
Nominal number of occupants in the room, NO	Discrete value from 0 to 4	-

**Table 3 ijerph-16-04349-t003:** List of input variables used for the TA prediction framework.

Classes	Interval
A	16 ≤ TA < 18
B	18 ≤ TA < 20
C	20 ≤ TA < 22
D	22 ≤ TA < 24
E	24 ≤ TA < 26
F	26 ≤ TA < 28
G	28 ≤ TA ≤ 30

**Table 4 ijerph-16-04349-t004:** Summary of the weather data parameters used in the building simulation model for Copenhagen (CPH), Athens (ATH) and Abu Dhabi (AUH). The format corresponds to: Minimum/First Quartile/Median/Third Quartile/Maximum.

Parameter	CPH	ATH	AUH
TAO, °C	−10/3/8/14/27	1/12/17/24/38	11/22/28/32/45
RHO, %	21/69/81/88/100	12/49/61/73/100	7/49/64/78/100
SR, W/m^2^	0/0/3/208/960	0/0/75/487/951	0/0/44/532/986

**Table 5 ijerph-16-04349-t005:** Main descriptive statistical parameters of the monitored variables used as input parameters for the TA prediction algorithm. STD: Standard deviation. DN: Dwelling number.

Bedrooms (BR)								Living Rooms (LR)							
DN		TA	TAO	RHO	SR	TRV	CO_2_	DN		TA	TAO	RHO	SR	TRV	CO_2_
1	Mean	23	14	71	265	14	809	1	Mean	21	14	72	262	14	634
	Median	22	14	71	150	14	650		Median	21	14	72	155	14	618
	Min	17	0	30	0	3	441		Min	17	0	30	0	3	435
	Max	30	27	100	918	22	1670		Max	28	27	100	918	22	1669
	STD	3	5	19	286	3	350		STD	2	5	19	283	3	139
2	Mean	22	8	75	207	8	1089	3	Mean	24	9	75	213	9	643
	Median	22	8	75	69	8	1017		Median	24	9	76	83	10	628
	Min	19	−7	28	0	−3	438		Min	22	−7	28	0	−3	448
	Max	26	24	100	904	19	3065		Max	28	26	100	906	21	1798
	STD	1	6	19	256	5	522		STD	1	6	19	258	5	128
3	Mean	19	4	86	96	4	674	4	Mean	25	9	69	206	9	1002
	Median	19	5	91	0	5	707		Median	24	9	71	68	9	989
	Min	17	−7	42	0	−3	461		Min	21	−5	25	0	−3	441
	Max	21	13	100	717	10	1140		Max	28	24	98	904	20	2149
	STD	1	4	15	161	3	119		STD	1	6	19	256	5	275
4	Mean	23	9	69	210	9	1353	5	Mean	21	9	80	113	10	594
	Median	23	9	70	75	9	1209		Median	21	9	84	4	11	584
	Min	18	−5	25	0	−2	446		Min	18	−7	30	0	−3	435
	Max	26	24	98	904	20	3634		Max	27	29	100	904	24	1280
	STD	1	6	19	257	5	650		STD	1	6	17	185	6	104
6	Mean	23	10	70	169	11	601	6	Mean	23	8	74	126	9	674
	Median	23	10	72	30	11	530		Median	22	7	77	7	8	615
	Min	19	−4	29	0	−1	438		Min	19	−4	31	0	−1	461
	Max	28	26	98	904	21	2913		Max	29	21	98	872	21	2965
	STD	2	5	19	236	4	216		STD	2	4	18	199	4	241
								7	Mean	22	8	71	167	9	558
									Median	22	8	74	21	8	543
									Min	20	−5	27	0	−3	450
									Max	25	26	98	904	20	1026
									STD	1	6	20	242	5	89

**Table 6 ijerph-16-04349-t006:** Information Gain and Pearson’s correlation coefficient for all input parameters, which were also ranked based on each index.

Attribute	Information Gain		Correlation (Pearson)	
	Value	Rank	Value	Rank
TRV	0.55	1	0.14	1
NO	0.10	6	0.10	6
FA	0.23	2	0.13	3
CY	0.15	4	0.03	8
WO	0.03	8	0.11	5
TRM	0.20	3	0.14	2
SR	0.12	5	0.05	7
RHO	0.09	7	0.11	4

**Table 7 ijerph-16-04349-t007:** Overall predictive performance of the method depending on the input parameters considered. “All” refers to all eight attributes; The prefix “w/o” means that this particular attribute was not included as input (e.g., w/o CY: all the attributes were considered as input except CY); The “Optimal case” included all attributes that decreased the performance when omitted, i.e., those that contributed to the predictive performance (highlighted cells). Results given in % of F1-score.

Input	CPH (Real)	CPH (Simulation)	ATH (Simulation)	AUH (Simulation)
All	29.8%	14.6%	25.8%	0.7%
w/o WO	29.5%	16.9%	26.1%	0.7%
w/o RHO	29.6%	15.1%	23.8%	0.6%
w/o SR	29.7%	14.8%	24.4%	0.7%
w/o TRV	30.7%	23.7%	24.7%	0.7%
w/o TRM	25.4%	11.7%	11.6%	0.7%
w/o CY	31.1%	10.6%	22.9%	6.8%
w/o FA	27.3%	21.3%	26.9%	0.7%
w/o NO	24.3%	9.2%	21.5%	9.4%
Optimal case	30.0%	24.8%	26.9%	13.3%

**Table 8 ijerph-16-04349-t008:** Estimation of the Predicted Mean Vote index (PMV) depending on the TA interval taken into account.

TA Interval	PMV (clo = 1), -	PMV (clo = 0.5), -	F1-score (CPH/ATH/AUH), %
16 ≤ TA < 18	−1.0 ± 0.2	−2.4 ± 0.3	0/0/0
18 ≤ TA < 20	−0.6 ± 0.2	−1.7 ± 0.3	50/2/0
20 ≤ TA < 22	−0.1 ± 0.2	−1.1 ± 0.3	35/68/0
22 ≤ TA < 24	0.3 ± 0.2	−0.5 ± 0.3	9/43/19
24 ≤ TA < 26	0.8 ± 0.2	0.1 ± 0.3	6/20/49
26 ≤ TA < 28	1.2 ± 0.2	0.7 ± 0.3	0/11/28
28 ≤ TA ≤ 30	1.7 ± 0.2	1.3 ± 0.3	0/1/31

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
