# Peer review of "Prediction of Indoor Air Temperature Using Weather Data and Simple Building Descriptors"

_ijerph, 2019, doi:10.3390/ijerph16224349_

Round 1

Reviewer 1 Report

It's not often that a paper is submitted for which the reviewers need only apply a light touch, but this is one of them and I thank the authors for putting in the effort to produce an excellent product. Nonetheless I still have some suggestions that will polish the paper somewhat.

Section 2.1: data collection.

I noticed none of your references used solar radiation as inputs to prediction models, but was glad to see your work did. A few references would be good here:

White-Newsone, J. , Brixa Sanchez, Olivier Jokkiet, Zhenzhen Zhang, Edith Parker, Timothy Dvonch, Marie O'Neill, 2012: Climate Change and Health: Indoor Heat exposure in Vulnerable Populations. Environ. Research, 112, 20-27.

Vant-Hull, B., P. Ramamurthy, B. Havlik, C. Jusino, C. Corbin-Mark, J. Keefe, M. Schuerman, J. Kumari-Drapkin, A. Glenn, 2018: The Harlem Heat Project: A Unique Media/Community Collaboration to Study Indoor Heat Waves. Bull. Amer. Meteor. Assc., 99, 2491-2506.

Section 2.3: Model Structure

I worry about a decision tree algorithm in this case because a dwelling has thermal inertia, and previous temperatures therefore affect current temperatures. This is largely addressed by your running mean temperatures, but several researchers have noticed a time lag of a day or more between indoor and outdoor temperatures, and you only used 3 hours. This may be a difference between short term and long term calculations: see Vant-Hull et all above that noted a 2 hour lag if hourly averages are used, but a 1 day lag when daily averages are used.

Section 2.4: Building Simulation Model

I was initially confused because actual data was collected, but this section claims a model is needed because it is difficult to collect data. I think the opening sentence should be framed differently to state that it is difficult to collect data across widely varying environments, so a dynamic building model was used to simulate these other environments. I'm glad to see some simple model of human environmental adjustment was applied but would like to see more details, since this is such an important input.

Sections 3.2,3: Performance evaluations

I have to say that since temperature is in reality a continuous variable, all the evaluations based on categories rather than numerical difference just comes across as noise. All people will care about is how far off the temperature is, and various measures of that. I'd dispense with all the category measures and lose several pages of evaluation. Much easier to read.

4. Discussion

Dividing into unequal bins really skews the categorical evaluation, another reason to dispense with it. Obviously the categories were needed for training, but the readers will not care to see all these measures unless converted to differences from temperature.

Author Response

We appreciate the constructive comments from the reviewer. We have attempted to answer them as follows:

It's not often that a paper is submitted for which the reviewers need only apply a light touch, but this is one of them and I thank the authors for putting in the effort to produce an excellent product. Nonetheless, I still have some suggestions that will polish the paper somewhat.

Comment #1: Section 2.1: data collection.

I noticed none of your references used solar radiation as inputs to prediction models, but was glad to see your work did. A few references would be good here:

White-Newsone, J. , Brixa Sanchez, Olivier Jokkiet, Zhenzhen Zhang, Edith Parker, Timothy Dvonch, Marie O'Neill, 2012: Climate Change and Health: Indoor Heat exposure in Vulnerable Populations. Environ. Research, 112, 20-27.

Vant-Hull, B., P. Ramamurthy, B. Havlik, C. Jusino, C. Corbin-Mark, J. Keefe, M. Schuerman, J. Kumari-Drapkin, A. Glenn, 2018: The Harlem Heat Project: A Unique Media/Community Collaboration to Study Indoor Heat Waves. Bull. Amer. Meteor. Assc., 99, 2491-2506.

Reply:

The references were added based on the suggestion from the reviewer on Page 2, line 77 and on Page 15, line 470. 

Comment #2: Section 2.3: Model Structure

I worry about a decision tree algorithm in this case because a dwelling has thermal inertia, and previous temperatures therefore affect current temperatures. This is largely addressed by your running mean temperatures, but several researchers have noticed a time lag of a day or more between indoor and outdoor temperatures, and you only used 3 hours. This may be a difference between short term and long term calculations: see Vant-Hull et all above that noted a 2 hour lag if hourly averages are used, but a 1 day lag when daily averages are used.

Reply:

The running mean temperature used in this study considered a period of 3 hours since in this point the correlation between indoor and outdoor temperatures was the highest with a Pearson correlation coefficient of 0.14 (Page 4, line 145). As suggested by the reviewer, a reference to Vant-Hull et al. was added on Page 15, line 470. The differences between building codes considered in the dwellings from our study and the study from Vant-Hull may have explained the discrepancies between the lag periods from both studies. 

Comment #3: Section 2.4: Building Simulation Model

I was initially confused because actual data was collected, but this section claims a model is needed because it is difficult to collect data. I think the opening sentence should be framed differently to state that it is difficult to collect data across widely varying environments, so a dynamic building model was used to simulate these other environments. I'm glad to see some simple model of human environmental adjustment was applied but would like to see more details, since this is such an important input.

Reply:

As suggested, a better explanation of why a building simulation model was applied was added on Page 6, line 212. Moreover, more details about the window adjustment approach were added on Page 7, line 239.

Comment #4: Sections 3.2,3: Performance evaluations

I have to say that since temperature is in reality a continuous variable, all the evaluations based on categories rather than numerical difference just comes across as noise. All people will care about is how far off the temperature is, and various measures of that. I'd dispense with all the category measures and lose several pages of evaluation. Much easier to read.

Reply:

The method presented in the manuscript considered a categorization of indoor temperatures in order to use a simple decision tree model that allows using multiple categorical and numerical input variables. The application of this prediction method in the app was not to indicate high accuracy, but merely to provide a quick and simple estimation of the indoor temperature to calculate PMV, also with expected reasonable accuracy. Section 3.2 presented a comprehensive analysis of the performance of the method when predicting each temperature class based on measured and simulated data. The actual performance of the algorithm will be presented in a more simple and general manner to the App users, accounting only for the overall efficacy of the method. Users will not be presented to a temperature category, but a PMV value estimated based on the center point of the estimated category. A brief explanation was added on Page 13, line 392.       

Comment #5: 4. Discussion

Dividing into unequal bins really skews the categorical evaluation, another reason to dispense with it. Obviously the categories were needed for training, but the readers will not care to see all these measures unless converted to differences from temperature.

Reply:

As mentioned by the reviewer, the discretization of the indoor temperature values had an effect on the performance of the method, which was discussed on Page 13, line 407. The app users will not meet this technical app functionality, but will be presented to an estimated PMV (in fact an index derived from PMV and shown on a gauge in the app dashboard). The manuscript describes the reasons for using a classification algorithm and the analysis of its performance under different conditions. The authors attempted to provide enough information for the understanding of the method and its potential improvement in the future.

We appreciate all comments from the reviewer, but in particular comments #4 and #5 as they bring up the importance of communicating the app output. These comments will be valuable when we further develop the app and its user interface.

Reviewer 2 Report

The manuscript proposes a method to predict indoor temperatures in houses based on outdoor weather and building descriptors. The manuscript is well-written and well-structured. The proposed method is to be implemented in a Smartphone App. To be accepted, the manuscript requires the following revisions.

To the reviewer, the practical application of this method is not clear. The reviewer recommends  that the authors describe some relevant use-case scenarios to demonstrate how this method can be applied in practice. Is the method intended to apply to any season? From the introduction, the main concern are extreme weather events. However, thermostat setting is a variable in this study, meaning that the buildings are mechanically heated and cooled, and therefore should provide acceptable conditions for comfort unless the systems do not have enough capacity. It can be argued that occupants' behaviors at home also depend on the season.

The explanation of the methodology is hard to follow. The reviewer suggests that the authors include a Figure to describe the methodology upfront: i.e. test data used to train the TA-prediction method, building simulation model data used as testing data, data and model used for performance evaluation model, etc. How and what data is used/processed between models?

The predictive performance of the indoor temperature prediction model seems to be accurate when applied to dwelling similar to the ones tested and under similar contextual factors. If this is the case, then the power of generality of the model is very low, and its applicability in real life is limited. If this is true, the authors should explain the limitations of the model, and further work needed to increase its generality.

The number of descriptors seem arbitrarily selected and small, and therefore the models have to rely on too many assumptions, which limit their power of predictability. How were the building descriptors selected? Would having more descriptors help improve the accuracy of the prediction models?  

Author Response

We appreciate the constructive comments from the reviewer. We have attempted to answer them as follows:

The manuscript proposes a method to predict indoor temperatures in houses based on outdoor weather and building descriptors. The manuscript is well-written and well-structured. The proposed method is to be implemented in a Smartphone App. To be accepted, the manuscript requires the following revisions.

Comment #1: To the reviewer, the practical application of this method is not clear. The reviewer recommends  that the authors describe some relevant use-case scenarios to demonstrate how this method can be applied in practice. Is the method intended to apply to any season? From the introduction, the main concern are extreme weather events. However, thermostat setting is a variable in this study, meaning that the buildings are mechanically heated and cooled, and therefore should provide acceptable conditions for comfort unless the systems do not have enough capacity. It can be argued that occupants' behaviors at home also depend on the season.

Reply:

The method presented in this study was designed as a framework to evaluate indoor environments, integrated in a smartphone App (Page 2, line 94). A better explanation of the use of the method was added around Page 3, line 99, based on the comment from the reviewer. The explanation reads:

“It is expected that the indoor module of the app will be particularly useful in environments hosting fragile individuals, such as young children or the elderly, and where heat or cold spells may result in unusual thermal exposures. Under these conditions, the app output may assist in an evaluation of coping actions.”

Furthermore, an idea that could be used for future research efforts was suggested on Page 16, line 532.

The method was not intended to be applied in any particular season. The data collected from field studies used to construct and validate the method was obtained from different seasons (Page 3, line 130). The decision tree model applied in this study is able to discriminate under which range of set-points there is an effect of the heating/cooling equipment over the indoor temperatures. This can be applied for heating set-points (the TRV measured in the field study) or cooling set-points (not present in the field study). It was assumed that the water radiators were correctly designed and had enough capacity to meet the heating demand of each dwelling. It was less likely to predict the most extreme indoor temperature measurements (see Figure 5), even though extreme weather events did not happen during the field study (see Table 5). The lack of measured data on the lowest and highest temperature intervals had a significant influence on the predictions, as mentioned on Page 14, line 429.

As mentioned by the reviewer, occupants’ behaviors depend on the season and this study aimed to include window opening and thermostat setting behavior across seasons on a simple manner.          

Comment #2: The explanation of the methodology is hard to follow. The reviewer suggests that the authors include a Figure to describe the methodology upfront: i.e. test data used to train the TA-prediction method, building simulation model data used as testing data, data and model used for performance evaluation model, etc. How and what data is used/processed between models?

Reply:

As suggested, a better explanation of the methodology was added on Page 3, line 106. Moreover, a figure that describes the methodology was included (see Figure 1). The data applied to construct the TA-prediction model was the measured data from field studies. The data from building simulations was only used to test the model under different climate conditions.   

Comment #3: The predictive performance of the indoor temperature prediction model seems to be accurate when applied to dwelling similar to the ones tested and under similar contextual factors. If this is the case, then the power of generality of the model is very low, and its applicability in real life is limited. If this is true, the authors should explain the limitations of the model, and further work needed to increase its generality.

Reply:

As mentioned by the reviewer, the applicability of the method depends on its capacity to predict temperatures under different conditions. The evaluation presented in the manuscript shows that the model had an overall accuracy of approximately 30% when predicting temperature categories, considering different dwellings and the same climate. However, it is challenging to obtain a comprehensive dataset that accounts for different climates and building types, which contains building-descriptors, occupant behavior parameters as well as indoor temperature values measured continuously during a long period. The limitations of the approach and future studies required in this topic were included on Page 16, line 537 and on Page 16, line 556.

Comment #4: The number of descriptors seem arbitrarily selected and small, and therefore the models have to rely on too many assumptions, which limit their power of predictability. How were the building descriptors selected? Would having more descriptors help improve the accuracy of the prediction models? 

Reply:

Based on the comment from the reviewer, an additional discussion about this issue was added in the manuscript on Page 14, line 440. The parameters were selected based on the literature review and the measured data available from previous field studies [27,28]. Even though eight parameters were used as inputs, the results from Table 6 showed that not all of them had the same impact on the performance of the method. The utilization of more descriptors has the potential to improve the performance of the method. Nonetheless, this will probably make the task of getting input from occupants more difficult and may induce over-fitting.

Reviewer 3 Report

The article is well written and organized.

Few inputs can enrich it.

Where in the scentific background do the authors place their study in this reverse engineering problem? See the outcom of the recent IAQVEC conference such as in https://iopscience.iop.org/article/10.1088/1757-899X/609/7/072043 or in https://iopscience.iop.org/article/10.1088/1757-899X/609/7/072022

In the conclusion, further explain the limitations of this approach.

Author Response

We appreciate the constructive comments from the reviewer. We have attempted to answer them as follows:

Comment #1: Where in the scientific background do the authors place their study in this reverse engineering problem? See the outcome of the recent IAQVEC conference such as in https://iopscience.iop.org/article/10.1088/1757-899X/609/7/072043 or in https://iopscience.iop.org/article/10.1088/1757-899X/609/7/072022

Reply:

Based on the suggestion from the reviewer, an additional discussion was added in the manuscript on Page 16, line 533. The method developed in this study could be used in future research studies to develop techniques to monitor indoor environments with fewer or no sensors. The importance of limiting the number of sensors was highlighted by Tronchin et al. [51], suggested by the reviewer.

Comment #2: In the conclusion, further explain the limitations of this approach.

Reply:

As suggested, an additional explanation of the limitations of the approach was included on Page 16, line 537 and on Page 16, line 556.